# High Tumor Mutation Burden Is Associated with Poor Clinical Outcome in EGFR-Mutated Lung Adenocarcinomas Treated with Targeted Therapy

**DOI:** 10.3390/biomedicines10092109

**Published:** 2022-08-29

**Authors:** Ji-Youn Sung, Dong-Won Park, Seung-Hyeun Lee

**Affiliations:** 1Department of Pathology, College of Medicine, Kyung Hee University, Seoul 02447, Korea; 2Division of Pulmonary Medicine and Allergy, Department Internal Medicine, College of Medicine, Hanyang University, Seoul 04763, Korea; 3Division of Pulmonary, Allergy, and Critical Care Medicine, Department of Internal Medicine, College of Medicine, Kyung Hee University, Seoul 02447, Korea

**Keywords:** lung cancer, next-generation sequencing, tumor mutation burden, survival, epidermal growth factor receptor, targeted therapy

## Abstract

**Simple Summary:**

Tumor mutation burden (TMB) represents the mutational load in the tumor cell genome and serves as a surrogate marker for tumor neoantigen production and potential immunogenicity. TMB has been widely explored as a complementary or alternative biomarker for immune checkpoint inhibitors in various malignancies including lung cancer. However, its clinical implication in the targeted-therapy setting is scarcely investigated. This study demonstrates that high TMB is independently associated with poor progression-free and overall survivals, and a low frequency of secondary T790M mutation in patients with epidermal growth factor receptor (*EGFR*)-mutated lung adenocarcinoma that are treated with tyrosine kinase inhibitors (TKIs). To the best of our knowledge, this is the first study demonstrating the significant association between the TMB level and resistance pattern after TKI failure.

**Abstract:**

This study aimed to determine the association between TMB and treatment outcomes in patients with epidermal growth factor receptor (EGFR)-mutated lung cancer that were treated with tyrosine kinase inhibitors (TKIs). The TMB was assessed using a 409-gene targeted next-generation sequencing panel. We compared the response rate (RR), progression-free survival (PFS), overall survival (OS), and frequency of secondary T790M mutations among the different TMB groups. The median TMB of the study population (n = 88) was 3.36/megabases. We divided 52 (59%) and 36 (41%) patients into the low and high TMB groups, respectively. A high TMB level was significantly associated with liver metastasis and more advanced stage (all *p* < 0.05). RR was significantly lower in the high TMB group than that of the low TMB group (50.0% vs. 80.7%, all *p* = 0.0384). In multivariate analysis, high TMB was independently associated with a shorter PFS (hazard ratio [HR] = 1.80, *p* = 0.0427) and shorter OS (HR = 2.05, *p* = 0.0397) than that of the low TMB group. Further, high TMB was independently associated with decreased T790M mutation development. These results suggest that high TMB may be a predictive biomarker for adverse treatment outcomes and represent a patients’ subgroup warranting tailored therapeutic approaches.

## 1. Introduction

Tyrosine kinase inhibitors (TKIs) targeting epidermal growth factor receptor (EGFR) have revolutionized the treatment landscape of the non-small cell lung cancers (NSCLCs) which harbor *EGFR* sensitizing mutations over the past decades. First-generation *EGFR*-TKIs including gefitinib and erlotinib have shown better efficacies over platinum-based doublet, and second-generation TKIs including afatinib and dacomitinib, subsequently proved their efficacies which are comparable or superior to those of the first-generation TKIs [1,2,3]. Nevertheless, a wide variation exists in the degree and duration of response to these TKIs among patients, and most of the *EGFR*-mutant tumors inevitably progress within 8 to 14 months due to the emergence of resistance [4,5]. Studies have revealed that acquired resistance can occur either in *EGFR*-dependent or EGFR-independent manner; the former includes the acquisition of *EGFR* T790M or C797X mutations, and the latter includes bypass track activation and other driving alterations such as *MET* amplification or histologic transformation [6]. Another breakthrough was made when novel third-generation TKIs were used to overcome the *EGFR* T790M-mediated resistance. In addition, osimertinib, a third-generation TKI, is now the standard of care for first-line treatment of patients with NSCLC harboring common *EGFR* mutations [7]. Hence, understanding the biology of resistance is critical for the development of new therapeutic strategies and improving the survival of the patients.

The advances in the molecular profiling techniques such as next-generation sequencing (NGS) have made a significant contribution toward current cancer precision medicine including the discovery of novel biomarkers and the identification of resistance mechanisms and potential therapeutic targets [8]. Tumor mutation burden (TMB), also called as tumor mutational load, denotes the number of acquired somatic mutations in the coding region of the cancer cell genome [9]. It has been widely explored as a complementary or alternative biomarker for immune checkpoint inhibitors (ICIs) in various malignancies including lung cancer [10,11,12,13]. High TMB refers to a high probability of tumor neoantigen production, and therefore, the likelihood of immune recognition and clearance by effector T-cells [14,15]. Meanwhile, in the context of targeted therapy, elevated TMB may be related to the increased possibility of resistance mutations or subclones harboring different molecular characteristics, which is related to resistance to drugs. However, the clinical implication of TMB in this treatment setting is largely unknown.

In this study, we aimed to determine the possible predictive or prognostic value of TMB in patients with *EGFR*-mutant lung adenocarcinoma that were treated with first-line TKIs. We also investigated the associations of TMB levels with the emergence of secondary T790M mutations after TKI failure.

## 2. Materials and Methods

### 2.1. Study Subjects and Data Collection 

We retrospectively enrolled patients with locally advanced or metastatic *EGFR* mutation-positive lung adenocarcinoma who received *EGFR*-TKIs as the frontline therapy at Kyung Hee University Hospital and Hanyang University Hospital, referral hospitals in South Korea, from March 2016 to July 2020. Patients with insufficient survival data, history of other malignant tumors newly diagnosed within 5 years, or other oncogenic drivers, including anaplastic lymphoma kinase (*ALK*), and ROS proto-oncogene 1 (*ROS1*) fusions, and those with a prior history of radiation therapy or chemotherapy were excluded.

Chest computed tomography (CT) scan, brain magnetic resonance imaging, and ^18^F-fluorodeoxyglucose positron emission tomography-computed tomography were used for staging workup in all the patients. The eighth edition of the TNM staging system of lung cancer by the International Association for the Study of Lung Cancer (IASLC) were used for the clinical staging [16]. The treatment response was regularly assessed with chest CT scans after every two or three cycles of systemic treatment according to the Response Evaluation Criteria in Solid Tumors (RECIST) 1.1 [17]. Patients’ medical records were reviewed to collect medical or social history, demographics, and survival data. The study protocol was reviewed and approved by the Institutional Review Board of Kyung Hee University Hospital (KHUH 2019-06-030). Written informed consent from all the participants that were alive was obtained. This study was conducted in compliance with the Declaration of Helsinki.

### 2.2. Next-Generation Sequencing and Calculation of TMB

Formalin-fixed paraffin-embedded tumor tissues that were obtained before initial *EGFR*-TKI treatment were used for genomic profiling. The detailed methods of sample preparation, sequencing, quality assessment, and variant calling are described in the Appendix A. Briefly, DNA extraction from tissues was performed using RecoverAll^TM^ Multi-Sample Isolation Kit (Thermo Fisher Scientific, Waltham, MA, USA) according to the manufacturer’s instructions. TMB was assessed using a 409-gene targeted NGS panel (Oncomine^TM^ Tumor Mutation Load assay, Thermo Fisher Scientific). For NGS library preparation, 5–40 ng of DNA was used depending on the availability of the input material. The libraries were purified using Agencourt AmpureXP beads (Beckman Coulter, Brea, CA, USA) and quantified by qPCR using the Ion Universal Quantitation Kit (Thermo Fisher Scientific). 

TMB calculations were performed on Ion Reporter^TM^ Analysis Software v5.10 (IR) using the Oncomine^TM^ Tumor Mutation Load w2.0 workflow. The TMB was calculated by dividing the number of nonsynonymous (missense and nonsense) somatic single nucleotide variants and coding indels by the number of exonic bases with at least 60X coverage and expressed as the number of mutations per megabase (Mb) of genome. The TMB values were rounded to two decimal places.

### 2.3. EGFR Mutation Testing

*EGFR* mutation tests were performed using tumor tissues. Genomic DNA was extracted from formalin-fixed, paraffin-embedded, 5-µm-thick tissue sections using the High Pure Template Preparation Kit (Roche Applied Science, Mannheim, Germany). The extracted DNA was stored at −20 °C until analysis. PANAMutyper^TM^ (PANAGENE Inc., Daejeon, Korea), a PNA-Clamping-based *EGFR* mutation detection kit was used for the detection of *EGFR* mutations. The primer sets covered mutations or deletions spanning exons 18–21 of the genes encoding the tyrosine kinase domain of *EGFR*. The results were interpreted according to the manufacturer’s instructions.

### 2.4. Statistical Analyses

The cut-off value for differentiating between low and high TMB was defined as the point with the lowest *p*-value by the log-rank test for all the possible TMB values. The baseline characteristics of different groups were compared using the Chi-square test or Fisher’s exact test, as appropriate. The clinical outcomes, including the response rate (RR), progression-free survival (PFS), and overall survival (OS) were assessed. The RR was defined as the percentage of patients who achieved complete or partial response. PFS was defined as the period from the first day of treatment to disease progression or death. OS was defined as the interval from the first day of treatment to death from any cause. Data of patients without tumor recurrence or death were censored at the last follow-up. Correlations between the survival outcomes and clinicopathological parameters were estimated by univariate analysis using the log-rank test, followed by Cox proportional hazard regression analysis. Parameters with *p* values < 0.2 in the univariate analysis were included for the multivariate analysis. The Kaplan-Meier method was used to estimate the survival rates. *p* < 0.05 was considered as statistically significant. All analyses were performed using SPSS v.20.0 (IBM Corporation, Armonk, NY, USA).

## 3. Results

### 3.1. Clinicopathological Characteristics of Patients

During the study period, 735 patients were newly diagnosed with NSCLC, while 243 patients were diagnosed with advanced lung adenocarcinoma. Of these patients, 131 received frontline *EGFR*-TKIs for *EGFR*-positive diseases. A total of fifteen patients with insufficient survival data, ten with concomitant cancers, and three who received other cancer treatments before targeted therapy were excluded. There were fifteen patients that were further excluded because the sequencing quality of their samples did not meet the minimum requirements. Finally, 88 patients were included for the TMB calculation and subsequent analysis. The modalities that were used for tissue acquisition were bronchoscopic biopsy (n = 5), transbronchial lung biopsy (n = 10), endobronchial ultrasonography-guided transbronchial needle aspiration (n = 28), and percutaneous needle biopsy (n = 45). 

Table 1 shows the clinical characteristics of the study population. All were Korean, with a median age of 67 years (range, 40–89 years). A total of 47 (53.4%) patients were aged ≥70 years, 44 (50.0%) were women, and 26 (29.5%) were current or former smokers. A total of 70 (79.5%) patients had an Eastern Cooperative Oncology Group performance status (ECOG PS) of 0 or 1. There were 15 (17.0%) and 73 (83.0%) patients that had Stage III and IV diseases, respectively. Further, 21 (23.9%) had metastases involving three or more organs, while 28 (31.8%) and 14 (15.9%) patients had the brain or liver metastases, respectively. A total of 52 (59.1%) patients had exon 19 deletion (19del), 31 (35.2%) had L858R point mutation, and 5 (5.7%) had uncommon or compound mutations. There were 68 (77.2%) patients that received afatinib, while 20 (22.9%) were treated with gefitinib or erlotinib as a first-line therapy.

### 3.2. TMB and Molecular Landscape

Figure 1 shows the TMB distribution among the study population. The median TMB of the study population was 3.36/Mb (range: 0.0–19.32). Among the 88 patients, 62 (70.4%) patients had at least one concomitant genetic alteration coexisting with mutant *EGFR*. The frequencies of the top 10 mutations are presented in Appendix A. *TP53* mutations were the most common occurring in 45.5% (40/88) of the patients, while *PIK3CA* (n = 17, 19.3%), *CTNNB1* (n = 14, 15.9%), and *SMARCA4* (n = 12, 13.6%) mutations also occurred frequently, which is consistent with previous reports [18,19]. 

### 3.3. Association between Clinicopathological Parameters and TMB

To investigate which clinicopathological parameters were associated with the TMB, we compared the median TMB levels between the groups within each parameter. The median TMB level showed a non-significant increasing trend in patients with more advanced stages (*p* = 0.0543) and was significantly higher in patients with liver metastasis (*p* = 0.0030) (Appendix A). We subsequently compared the distribution of patients according to the TMB group. The optimal cut-off value for low and high TMB levels was determined to be 2.53/Mb by the log rank test. Using this cut-off, 52 (59.1%) and 36 (40.9%) patients were classified into the low and high TMB groups, respectively. As shown in Table 1, TMB was not related to parameters such as age, sex, smoking history, ECOG PS, and *EGFR* subtype. However, a high TMB level was significantly associated with a more advanced stage (*p* = 0.0171), three or more organs involvement (*p* = 0.0249), and liver metastasis (*p* = 0.0210). 

### 3.4. Distribution of Co-Mutations according to TMB Levels

To further examine whether any of the co-mutations were enriched in the high TMB group, we investigated the frequency of co-mutations at different TMB levels. *TP53* mutations were significantly more common in the high TMB group (23/36, 63.8%) than in the low TMB group (17/52, 32.6%; *p* =0.0254; Appendix A). *PIK3CA* (9/36, 25.0% vs. 8/52, 15.4%) and *CTNNB1* (7/36, 19.4% vs. 7/52, 13.4%) mutations were more frequent in the high TMB group, but the differences were not statistically significant.

### 3.5. Response Rate and PFS According to TMB Level

The median follow-up period was 38.9 months (range: 3.3–57.8 months). Table 2 summarizes the treatment response according to the TMB levels. In the low TMB group, three (5.8%) and 40 (76.9%) patients showed a complete response (CR) and partial response (PR), respectively, while 18 (50%) patients showed PR in the high TMB group. The RR was significantly lower in the high TMB group than that of the low TMB group (50.0% versus 80.7%, respectively, *p* = 0.0384).

Table 3 shows the PFS analysis results according to clinicopathological parameters. A total of 73 patients (82.9%) progressed during the follow-up period. The median PFS of the study population was 17.7 months (range, 3.1–44.7 months). Univariate analysis showed that male sex, three or more organ involvement, presence of liver metastasis, and positive *TP53* mutation were associated with poor PFS (all *p* < 0.05). In addition, a high TMB level was significantly associated with shorter PFS (*p* = 0.0205). Multivariate analysis showed that male sex (hazard ratio [HR] = 1.84, 95% confidence interval [CI]: 1.10–3.08), presence of liver metastasis (HR = 2.01, 95% CI: 1.06–5.77), and high TMB (HR = 1.80, 95% CI: 1.17–4.43) were independently associated with shorter PFS. Patients with high TMB levels were likely to have poor PFS compared to those with low TMB levels as shown in the Kaplan–Meier survival curves (Figure 2a).

### 3.6. OS According to TMB Level

Table 4 shows the results of the OS analysis. The median OS of the study population was 35.5 months (range, 3.1–54.8 months). Among 73 patients who progressed after first-line TKI therapy, 52 (71.2%) underwent T790M mutation testing, all of whom received second-line treatment. Overall, 22 patients (42.3%) harbored the T790M mutation. The 22 patients who progressed with T790M mutation after frontline EGFR-TKIs were treated with osimertinib, and 30 patients received other treatments, either pemetrexed alone (n = 5) or pemetrexed/platinum doublet (n = 23). Patients who underwent small cell lung cancer transformation received etoposide/platinum (n = 2). Detailed data on salvage treatments are provided in Appendix A. Univariate analysis showed that metastases involving three or more organs and the presence of liver metastasis were significantly associated with shorter OS (all *p* < 0.05). In addition, a high TMB level was also significantly associated with shorter OS (*p* = 0.0080). Multivariate analysis showed that the presence of liver metastasis (HR = 2.17, 95% CI: 1.14–4.57) and high TMB levels (HR = 2.05, 95% CI: 1.04–4.07) were independently associated with shorter OS. Patients with high TMB levels were likely to have poor OS compared to those with low TMB levels as shown in the Kaplan–Meier survival curves (Figure 2b).

### 3.7. Frequency of Acquired T790M Mutation According to TMB Levels

To identify whether the TMB level is associated with the development of a secondary T790M mutation, we evaluated the mutation rates according to different TMB groups. Among 73 patients who progressed after first-line TKI therapy, 52 (71.2%) underwent T790M mutation testing. Table 5 summarizes the frequencies of T790M mutation stratified by clinicopathological parameters of the 52 patients. The mutation rates were significantly lower in patients with the L858R mutation and those who received TKI for less than 12 months (all *p* < 0.05). In addition, patients with high TMB levels showed a significantly lower frequency of T790M mutation than that of those with low TMB levels (26.4.7% vs. 51.2%, respectively, *p* = 0.0377; Figure 3). Univariate analysis for the factors that were associated with the frequency of T790M mutation showed that the L858R mutation, TKI use for less than 12 months, and high TMB levels were significantly associated with a lower incidence of T790M mutation (all *p* < 0.05). Multivariate analysis showed that the L858R mutation (odds ratio [OR] = 0.46, 95% CI: 0.08–0.94), TKI use for less than 12 months (OR = 0.28, 95% CI: 0.14–0.85), and high TMB levels (OR = 0.42, 95% CI: 0.17–0.96) were independently associated with a low incidence of acquired T790M mutation. 

## 4. Discussion

Our study demonstrated that a high TMB level was independently associated with poor PFS and OS in patients with *EGFR*-mutated lung adenocarcinoma that were treated with frontline targeted therapy. In addition, TMB was also associated with a lower rate of secondary T790M mutations after TKI use. To the best of our knowledge, this is the first study demonstrating the significant association between the TMB level and resistance pattern after TKI failure.

The TMB level that was observed in the present study (median 3.36/Mb) was much lower than that which was observed in previous studies on unselective NSCLC [20,21] but was comparable to those of previous studies that reported low TMB levels in *EGFR*-mutant tumors [19,22]. In a study analyzing an MSK-IMPACT cohort (n = 1668 including 410 *EGFR*-mutant patients) from The Cancer Genome Atlas (TCGA) database, the median TMB values of wild-type *EGFR*, non-sensitive *EGFR* mutations, exon 19 deletions, and L858R were 6.12, 5.66, 3.77, and 4.72, respectively [22]. In that study, the authors also reported a similar trend in TMB levels in the Chinese cohort in each patient group (6.10, 4.95, 4.10, and 3.10, respectively, n = 292). Another study that was conducted using MSK-IMPACT NGS platform (n = 783 including 153 *EGFR*-mutant patients) reported a median TMB of 3.77/Mb in an *EGFR*-mutant population, which is significantly lower than that of *EGFR*-wild type [19]. Although we did not compare the TMB levels according to *EGFR* mutational status, our study confirmed previous findings of relatively low TMB levels in *EGFR*-positive tumors. Evidence indicates that ICIs are not effective in the oncogene-driven lung cancers, and it is partly due to an immunosuppressive tumor microenvironment (TME) in such tumors, including the recruitment of tumor-associated macrophages and regulatory T-cells and the production of inhibitory cytokines that are induced by the activated *EGFR* signaling [23]. TMB represents the mutational load in the tumor cell genome and serves as a surrogate marker for tumor neoantigen production and potential immunogenicity. Thus, based on the previous and our data, the low TMB can be another mechanism to explain such poor efficacy of immunotherapeutic approaches in *EGFR*-positive lung cancer.

The current data showed that *TP53* mutations were associated with short PFS in univariate analysis and were more enriched among those with high TMB. Mutations in TP53 can be found in 35–60% of NSCLC patients, more frequently in squamous cell carcinoma, and in smokers [24,25]. *TP53* mutations are associated with a poor response and shorter survival in patients with lung cancer that were treated with chemotherapy or surgical resection [26,27]. Similarly, accumulating evidence suggests that these mutations are also negative prognostic factors in *EGFR*-positive NSCLC [28,29,30]. The mechanism by which *TP53* alterations are associated with poor outcomes in these populations remains unclear. A recent study by Lee et al. demonstrated that mutant *TP53*-induced epithelial-to-mesenchymal transition (EMT)-mediated resistance, and *TP53* silencing led to primary resistance to EGFR-TKIs through AXL induction, suggesting that these mutations can be associated with both primary and acquired resistance to EGFR-TKIs [31]. In our data, the association between high TMB and poor clinical outcomes remained significant, even after adjusting for *TP53* mutational status. This suggests that high TMB is an independent predictive and prognostic factor in the *EGFR*-positive population, although the enrichment of certain co-mutations in patients with higher TMB could contribute to worse outcomes.

In the present study, we identified that TMB was associated with more advanced stage and liver metastasis. Cumulative evidence has demonstrated that liver metastasis is associated with poor clinical outcomes in patients that are receiving the first- and second-generation *EGFR*-TKIs [32,33,34]. In our study, patients with liver metastasis showed significantly shorter PFS and OS, consistent with those previous findings. A very recent real-world study comparing the efficacies of different EGFR-TKIs showed that osimertinib, a third-generation *EGFR*-TKI, was not superior over other TKIs in patients with liver metastasis, while it provided significant clinical benefits in patients with brain or bone metastasis [35]. The association between liver metastasis and poor prognosis has been consistently reported across different treatment settings, including immunotherapy [36]. The exact underlying mechanism of the dismal prognostic impact of such a type of metastasis in lung cancer has not been fully understood; however, it can be partly attributed to an activation of the bypass tracts and enhanced escape from immune surveillance [37,38,39]. It has been reported that insulin-like growth factor 1 (IGF-1), the ligand of the IGF-1 receptor (IGF-1R), is highly expressed in the TME of the liver metastasis, and the signaling from IGF-1R promotes the tolerance to osimertinib in *EGFR*-positive lung cancer [40]. In addition, the expression of vascular endothelial growth factor (VEGF) is increased in the liver metastasis compared with other metastatic sites, and VEGF and *EGFR* signaling share downstream pathways [37]. Thus, the up-regulated *EGFR* signaling in *EGFR*-mutant cells can activate VEGF signaling through hypoxia-independent mechanisms, which in turn results in the emergence of resistance to TKIs [37]. Notably, a clinical trial of the combinational effect of an *EGFR*-TKI and an anti-VEGF receptor antibody demonstrated superior efficacy over TKI alone in patients with liver metastases [38]. Moreover, lower CD8 + T-cell infiltration in both the primary tumors and extrahepatic metastatic lesions has been observed in patients with liver metastasis, and this suggest that liver metastasis can induce systemic immunosuppression [39,41]. There is no study evaluating the clinicopathological factors which might be related with TMB so far. The present data suggest the possible association between high TMB and liver metastasis, and the unfavorable prognosis in the high TMB patients may be partly attributed to the accelerated liver metastasis. Our findings should be validated by further investigations.

The relationship between high TMB and poor clinical outcome in *EGFR*-mutated NSCLC has been suggested in two independent studies [19,22]. In one study, TMB was significantly associated with a shorter time-to-treatment discontinuation and OS, and the TMB levels were increased at the time of disease progression when comparing the pretreatment and post-progression samples [19]. The other study also reported a negative predictive value of TMB using TCGA data and validated their results using Chinese cohort [22]. In the present study, TMB was associated with poor PFS and OS, which is consistent with those previous data, and the association was significant even after adjusting the effect of liver metastasis in the multivariate analysis. A growing number of studies have demonstrated that various genomic alterations, including the activation of oncogenes and inactivation of tumor suppressor genes, are related with organotropism in metastasis in lung cancer [42]. For example, MYC, YAP1, or MMP13 overexpression were shown to be associated with the incidence of brain metastasis [43]. However, data on such genomic association or metastatic driver for liver metastasis in lung cancer are very limited. A recent phylogenetic analysis using paired primary tumors and metastases of lung adenocarcinoma reported that the genetic profiles are highly similar between the primary lung lesions and liver metastasis, and tumor cells in liver metastasis are genetically diverged from those in the primary tumor at a relatively later stage compared with the brain metastasis [44]. Such findings indicate that the liver metastasis follows the linear progression model of metastasis rather than the parallel model. In that study, the TMB levels were comparable between the primary tumor and liver metastasis [44]. More recently, another group analyzed the relationship between the genomic features of metastatic cancers and their organ-specific patterns of metastasis (n = 21,546) and reported the significant associations between TMB and organ-specific patterns of metastasis in several tumor types [45]. In that study, high TMB was associated with (1) lung adenocarcinoma to the brain and adrenal gland, (2) pancreatic adenocarcinoma to the liver, and (3) head and neck squamous to head and neck cancer. Based on our and previous data, we hypothesize that the high burden of somatic mutations can be a metastatic driver of liver metastasis or can induce genetic instability to promote metastasis and tumor progression. Further investigations are needed to validate our hypothesis.

The identification of secondary T790M mutation is critical for the management of *EGFR*-mutant patients because T790M-positive resistance is readily responsive to third-generation EGFR-TKIs, including osimertinib and lazertinib [46,47]. Previous data have demonstrated that the association between the more frequent acquisition of T790M mutation in 19del and the long duration of treatment with *EGFR*-TKIs [48,49,50]. Our results confirmed these findings, and additionally suggest that high TMB level may be associated with less emergence of T790M after prior TKI failure. The association between TMB and the frequency of the T790M mutation has scarcely been evaluated. In a study, T790M-negative tumor at progression showed relatively high TMB compared with T790M-positive tumors [51]. Another study reported that the TMB level showed a non-significant trend of increase in T790M-negative resistance [19]. Similarly, we found a significant association between the TMB and the resistance pattern, and our data suggest that baseline high TMB may be a negative predictor of T790M-positive resistance. It is well known that T790M-positive resistance is associated with more favorable outcomes than T790M-negative resistance [5]. The low frequency of T790M mutations in the high-TMB group may explain the worse OS in these populations. The mechanism by which TMB is associated with less development of the resistance mutation is unclear; however, it may be explained by the different durations of treatment with *EGFR*-TKI according to different TMB levels. As suggested in the previous studies, T790M-positive cells can be enriched or newly emerged during TKI treatment [52,53,54]. In addition, substantial time is required for the development of T790M-dominant tumor because the cells harboring the mutation grow indolently compared to T790M-negative cells [55]. Taken together, we can hypothesize that the low frequency of T790M mutation can be attributed to the shorter duration of treatment in the patients with high TMB. 

The present data suggest that high TMB confers a distinct aggressive phenotype that may require different treatment strategies. As TKI monotherapy could be less effective in patients with a high TMB level, other therapeutic strategies, such as a combinational approach using chemotherapy or immunotherapy, might be feasible. Notably, a previous study showed that gefitinib plus platinum doublet showed better clinical outcomes in terms of PFS and OS compared with gefitinib alone in *EGFR*-mutant NSCLC [56]. Very recently, amivantamab, a bispecific EGFR-MET antibody, in combination with lazertinib, a third-generation EGFR-TKI, showed promising results in patients who are previously treated with osimertinib and platinum-based chemotherapy [57]. Prospective investigations are required to determine the optimal treatment strategy for those with different TMB levels. 

This study had several limitations. First, it was a relatively small retrospective study that was performed at only two institutes, and using a data-driven TMB cutoff value, which could have increased the risk of bias. Second, all the patients were treated with either the first-or second-generation *EGFR*-TKIs, and data on the use of frontline osimertinib were not available. Third, unlike the FoundationOne and MSK-IMPACT panel tests, which were approved by the FDA, the assay that was adopted in this study (Oncomine Tumor Mutational Load assay) has not yet been validated for clinical use. However, previous studies revealed good concordance with TMB assessment by whole exome sequencing [58,59], and a recent study that was conducted by the Onconetwork Immuno-Oncology Consortium showed that the platform that we used demonstrated robustness and reproducibility for TMB evaluation [60]. Fourth, data on TMB levels at the time of disease progression and blood TMB levels were not analyzed. The blood TMB, determined using circulating tumor DNA, is emerging as a useful biomarker in different anti-cancer treatment settings [61]. Thus, the dynamics of TMB levels in tumor tissues and in the blood during treatment could be an interesting topic for future studies. Finally, we did not simultaneously explore the immune phenotypes that could affect the TKI resistance. We are currently working on the possible impact of immunologic signatures on the prognosis of *EGFR*-mutant populations.

## 5. Conclusions

The present data demonstrate that a high TMB level was associated with dismal clinical outcomes, not only due to the poor response to frontline TKI treatment but also the less emergence of T790M-associated resistance. Although further studies are needed to verify our results, the current findings highlight that a high TMB level might confer an aggressive phenotype, requiring tailored therapeutic approaches among *EGFR*-mutant patients. Future investigations should focus on the determination of the optimal cutoff of TMB among different NSG platforms, the possible predictive value of TMB in other malignancies harboring driver oncogenes, and the optimal treatment strategy for the TMB-high patients. Large prospective studies may facilitate the clinical use of TMB for the prediction of prognosis and risk stratification.

## Figures and Tables

**Figure 1 biomedicines-10-02109-f001:**
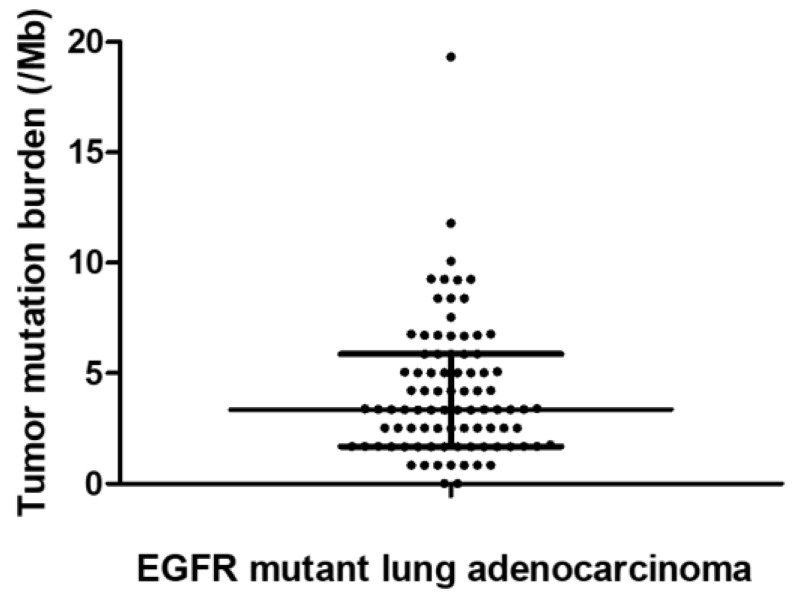
Distribution of TMB among the study population. Median TMB was 3.36/Mb. Bars indicate the median and interquartile range.

**Figure 2 biomedicines-10-02109-f002:**
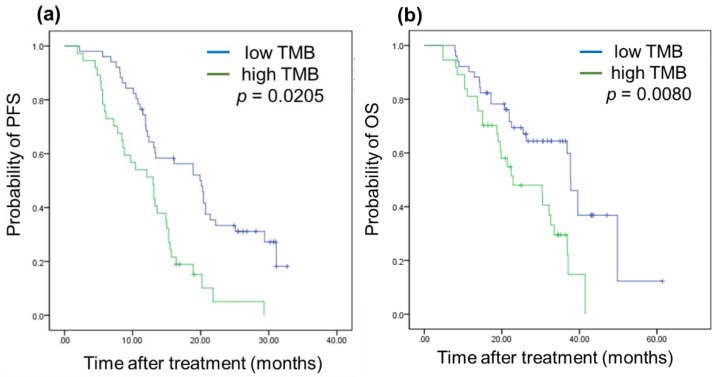
Kaplan-Meier curves of progression-free survival (PFS, (**a**)) and overall survival (OS, (**b**)) according to the different expression levels of TMB. *p*-values were determined using the log-rank test.

**Figure 3 biomedicines-10-02109-f003:**
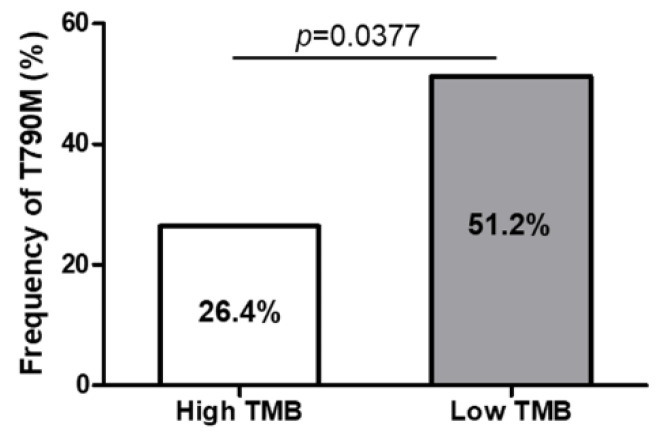
Frequency of secondary T790M mutation after tyrosine kinase inhibitor failure according to different TMB level. The rate of T790M mutation is significantly lower in the high TMB group compared with the low TMB group (*p* = 0.0377).

**Table 1 biomedicines-10-02109-t001:** Characteristics of the 88 study patients stratified by TMB levels.

	No. of Patients (%)	TMB		*p*-Value
		Low (≤2.53, n = 52)	High (>2.53, n = 36)	
Age				0.7370
<70	41 (46.6)	25 (48.1)	16 (44.4)	
≥70	47 (53.4)	27 (51.9)	20 (55.6)	
Sex				0.1933
Male	44 (50.0)	29 (55.8)	15 (41.7)	
Female	44 (50.0)	23 (44.2)	21 (58.3)	
Smoking history				0.1840
Never	62 (70.5)	35 (67.3)	27 (75.0)	
Ever	26 (29.5)	17 (32.7)	9 (25.0)	
Smoking intensity				0.7591
<30 pack-years	72 (81.8)	42 (80.8)	30 (83.3)	
≥30 pack-years	16 (18.2)	10 (19.2)	6 (16.7)	
ECOG PS				0.3791
0, 1	70 (79.5)	43 (82.7)	27 (75.0)	
≥2	18 (20.5)	9 (17.3)	9 (25.0)	
Stage				0.0171
III	15 (17.0)	13 (25.0)	2 (5.6)	
IV	73 (83.0)	39 (75.0)	34 (94.4)	
Involved organ				0.0249
<3	67 (76.1)	44 (84.6)	23 (63.9)	
≥3	21 (23.9)	8 (15.4)	13 (36.1)	
Brain metastasis				0.6532
No	60 (68.2)	35 (67.3)	25 (69.4)	
Yes	28 (31.8)	17 (32.7)	11 (30.6)	
Liver metastasis				0.0210
No	74 (84.1)	47 (90.3)	27 (75.0)	
Yes	14 (15.9)	5 (9.7)	9 (25.0)	
*EGFR* subtypes				0.1898
19del	52 (59.1)	29 (52.7)	22 (61.1)	
L858R	31 (35.2)	20 (38.4)	12 (33.3)	
Others	5 (5.7)	3 (5.7)	2 (5.5)	
First-line TKI				0.2451
Gefitinib	14 (15.9)	8 (15.3)	6 (16.7)	
Erlotinib	6 (6.8)	3 (5.7)	3 (8.3)	
Afatinib	68 (77.2)	41 (78.8)	27 (75.0)	

ECOG PS, Eastern Cooperative Oncology Group Performance Status; EGFR, epidermal growth factor receptor; 19del, deletion mutation at exon 19; TKI, tyrosine kinase inhibitor.

**Table 2 biomedicines-10-02109-t002:** Treatment response according to the different TMB levels.

	No. of Patients (%)		*p*-Value
	Low TMB (n = 52)	High TMB (n = 36)	
Response rate (CR + PR)	42 (80.7)	18 (50.0)	0.0384
CR	3 (5.8)	0 (0)	
PR	40 (76.9)	18 (50.0)	
SD	8 (15.4)	12 (33.3)	
PD	1 (1.9)	6 (16.7)	

CR, complete remission; PR, partial remission; SD, stable disease; PD, progressive disease.

**Table 3 biomedicines-10-02109-t003:** Progression-free survival analyses results according to clinicopathological parameters of all the study subjects.

	Median PFS (Months)	Univariate		Multivariate	
		HR (95% CI)	*p*-Value	HR (95% CI)	*p*-Value
All	17.7				
Age			0.4534	NA	
<70	18.9	reference			
≥70	16.6	1.2 (0.75–1.93)			
Sex			0.0323		0.0214
Male	15.1	1.68 (1.04–2.69)		1.83 (1.11–3.22)	
Female	20.9	reference		reference	
Smoking history			0.6920	NA	
Never	18.6	reference			
Ever	16.1	1.11 (0.66–1.87)			
Smoking intensity			0.1567		0.3650
<30 pack-years	18.3	reference		reference	
≥30 pack-years	13.8	1.55 (0.85–2.84)		1.26 (0.74–2.79)	
ECOG PS			0.3045		
0, 1	19.3	reference			
≥2	16.2	1.34 (0.77–2.36)			
Stage			0.1771		0.8555
III	19.3	reference		reference	
IV	16.4	1.57 (0.82–3.04)		1.21 (0.56–2.37)	
Involved organ			0.0172		0.3258
<3	19.5	reference		reference	
≥3	15.4	1.97 (1.13–3.43)		1.50 (0.78–2.98)	
Brain metastasis			0.6139	NA	
No	19.2	reference			
Yes	15.5	1.14 (0.69–1.87)			
Liver metastasis			0.0189		0.0371
No	19.1	reference		reference	
Yes	14.7	2.15 (1.46–5.19)		2.05 (1.09–5.86)	
*EGFR* subtypes *			0.2630	NA	
19del	18.7	reference			
L858R	17.1	1.23 (0.23–2.44)			
First-line TKI			0.0735		0.1321
Gefitinib/erlotinib	17.3	1.66 (0.97–4.16)		1.52 (0.89–6.73)	
Afatinib	20.6	reference		reference	
*TP53* mutation			0.0311		0.0894
Negative	19.1	reference		reference	
Positive	14.5	1.88 (1.12–3.09)		1.56 (0.26–4.15)	
TMB level			0.0205		0.0427
Low (<2.53)	20.1	reference		reference	
High (≥2.53)	13.4	1.98 (1.09–2.89)		1.80 (1.17–4.43)	

* Analysis for 83 patients excluding 5 patients with uncommon or compound mutations. ECOG PS, Eastern Cooperative Oncology Group Performance Status; EGFR, epidermal growth factor receptor; 19del, deletion mutation at exon 19; TKI, tyrosine kinase inhibitor; HR, hazard ratio; CI, confidence interval; NA, not analyzed.

**Table 4 biomedicines-10-02109-t004:** Overall survival analyses results according to the clinicopathological parameters of all the study subjects.

	Median OS (Months)	Univariate		Multivariate	
		HR (95% CI)	*p*-Value	HR (95% CI)	* *p*-Value
All	35.5				
Age			0.7403	NA	
<70	36.8	reference			
≥70	32.2	1.12 (0.47–1.99)			
Sex			0.4781	NA	
Male	32.6	1.22 (0.70–2.13)			
Female	36.9	reference			
Smoking history			0.4489	NA	
Never	36.8	reference			
Ever	32.2	1.26 (0.69–2.31)			
Smoking intensity			0.5614	NA	
<30 pack-years	36.8	reference			
≥30 pack-years	32.6	1.24 (0.60–2.55)			
ECOG PS			0.6378	NA	
0, 1	37.7	reference			
≥2	32.5	1.15 (0.73–2.22)			
Stage			0.1443		0.5604
III	37.5	reference		reference	
IV	32.2	2.01 (0.79–5.04)		1.35 (0.49–3.76)	
Involved organ			0.0346		0.4893
<3	36.9	reference		reference	
≥3	21.4	1.94 (1.05–3.57)		1.28 (0.64–2.56)	
Brain metastasis			0.1690		0.3589
No	37.1	reference		1.35 (0.71–2.59)	
Yes	30.4	1.53 (0.83–2.81)		reference	
Liver metastasis			0.0097		0.0314
No	36.9	reference		Reference	
Yes	21.9	2.34 (1.25–4.40)		2.17 (1.14–4.57)	
*EGFR* subtypes *			0.2154	NA	
19del	36.9	reference			
L858R	30.8	1.56 (0.36–6.73)			
First-line TKIs					0.2019
Gefitinib/erlotinib	30.2	1.70 (0.95–3.57)	0.0847	1.41 (0.69–3.27)	
Afatinib	36.1	reference		reference	
*TP53* mutation			0.3142	NA	
Negative	35.7	reference			
Positive	30.3	1.47 (0.89–5.23)			
TMB level			0.0080		0.0397
Low (<2.53)	37.1	reference		reference	
High (≥2.53)	21.9	2.65 (1.50–4.67)		2.05 (1.04–4.07)	

* Analysis for 83 patients excluding 5 patients with uncommon or compound mutations. ECOG PS, Eastern Cooperative Oncology Group Performance Status; EGFR, epidermal growth factor receptor; 19del, deletion mutation at exon 19; TKI, tyrosine kinase inhibitor; HR, hazard ratio; CI, confidence interval; NA, not analyzed.

**Table 5 biomedicines-10-02109-t005:** Analysis for the factors that were associated with the emergence of secondary T790M mutation (n = 52).

	T790M		*p*-Value *	Univariate Analysis		Multivariate Analysis	
	Negative (n = 30)	Positive (n- = 22)		OR (95% CI)	*p*-Value	OR (95% CI)	*p*-Value
Age			0.3710		0.2513	NA	
<70	15 (55.5)	12 (45.5)		0.62 (0.12–1.87)			
≥70	15 (40.0)	10 (60.0)		reference			
Sex			0.5956		0.6947	NA	
Male	19 (65.5)	10 (34.5)		0.82 (0.11–2.23)			
Female	11 (47.8)	12 (52.2)		reference			
Smoking history			0.1781		0.2413	NA	
Never	19 (54.2)	16 (45.8)		0.86 (0.23–4.57)			
Ever	11 (64.7)	6 (35.3)		reference			
Smoking intensity			0.3124		0.2991	NA	
<30 pack-years	18 (51.4)	17 (48.6)		0.75 (0.13–3.46)			
≥30 pack-years	12 (70.5)	5 (29.6)		reference			
ECOG PS			0.4136		0.5660	NA	
0, 1	16 (42.1)	12 (57.9)		reference			
≥2	14 (58.3)	10 (41.7)		0.41 (0.23–3.62)			
Stage			0.5480		0.9608	NA	
III	4 (44.4)	5 (55.6)		reference			
IV	26 (60.4)	17 (39.6)		0.78 (0.37–4.95)			
Involved organ			0.9618		0.8068	NA	
<3	20 (58.8)	14 (41.2)		reference			
≥3	10 (55.5)	8 (44.5)		0.95 (0.32–3.57)			
Brain metastasis			0.2176		0.1425		0.9816
No	20 (64.5)	11 (35.5)		reference		reference	
Yes	10 (47.6)	11 (52.4)		0.66 (0.17–3.05)		0.98 (0.25–3.89)	
Liver metastasis			0.6902		0.5346	NA	
No	22 (59.5)	17 (40.5)		reference			
Yes	8 (61.5)	5 (38.5)		0.70 (0.22–3.39)			
*EGFR* subtypes			0.0471		0.0334		0.0406
19del	11 (50.0)	11 (50.0)		reference		reference	
L858R	19 (63.3)	11 (36.7)		0.38 (0.17–0.99)		0.46 (0.08–0.94)	
First-line TKIs *			0.8460		0.6291	NA	
Gefitinib/erlotinib	11 (55.0)	8 (45.0)		reference			
Afatinib	19 (57.7)	14 (42.3)		0.75 (0.51–2.78)			
Duration of TKI use			0.0324		0.0397		0.0423
<12 months	15 (75.0)	5 (25.0)		0.32 (0.10–0.91)		0.28 (0.14–0.85)	
≥12months	15 (46.8)	17 (53.2)		reference		reference	
TMB expression			0.0377		0.0416		0.0479
Low (<2.53)	16 (48.8)	17 (51.2)		reference		reference	
High (≥2.52)	14 (73.6)	5 (26.4)		0.30 (0.07–0.88)		0.42 (0.17–0.96)	

* Analysis after excluding 2 patients with uncommon or compound mutations. ECOG PS, Eastern Cooperative Oncology Group Performance Status; EGFR, epidermal growth factor receptor; 19del, deletion mutation at exon 19; TKI, tyrosine kinase inhibitor; OR, odds ratio; CI, confidence interval; NA, not analyzed.

## Data Availability

Data are available on request due to privacy and ethical restrictions.

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
