# Peer review of "High Tumor Mutation Burden Is Associated with Poor Clinical Outcome in EGFR-Mutated Lung Adenocarcinomas Treated with Targeted Therapy"

_biomedicines, 2022, doi:10.3390/biomedicines10092109_

Round 1
Reviewer 1 Report
This is a retrospective observational study of patients with EGFR mutated NSCLC treated with 1st-2nd gen TKIs, in which the authors aimed to determine the association between TMB and treatment outcomes. This study was designed with a descriptive aim, without a hypothesis-driven fixed sample size. TMB was assessed by means of Thermo Fisher Scientific Oncomine Tumor Mutation Load Assay.
With 88 patients included, median TMB was 3.36/Mb. Investigators identified a data-driven cutoff value for low and high TMB levels (2.53/Mb by the log rank test). 41% patients had higher TMB levels and a worse outcome in terms of ORR, PFS and OS. They also found a lower incidence of acquired T790M mut in this group.
The paper is well written and of academic interest. I would suggest some revisions:
1) Introduction: the authors declare “third-generation TKIs have been used to overcome the EGFR T790M-mediated resistance”. In addition, I would state that Osimertinib, a third gen TKI, is now the standard of care for first line treatment of patients with NSCLC harboring EGFR common mutations.
2) Results: the authors do not provide data about co-mutations in lower and higher-TMB populations. For example, the presence of TP53 mutations (the most frequent mutations found in the whole population of this study according to Fig. S1) is a well-known negative prognostic factor in EGFR mutated NSCLC patients (Aggarwal et al., JCO PO 2018. Jiao et al., Lung Cancer 2018). The enrichment of such mutations in patients with higher TMB could contribute to explain the worse outcome.
3) Table 3: presence of brain mets appears as reference for HR calculation. Is it a compilation error?
4) Discussion: this section could be shortened, but I leave the decision to the authors.
I. Limitations: this section should be improved:
a) I would point out the limitations regarding the assessment of TMB by NGS. Gene panels differ in input sample requirement, gene number or identity, region covered, workflow, and bioinformatic algorithms used. While FoundationOne and MSK-IMPACT panel test have documented good concordance with TMB assessment by WES and are approved by FDA, the assay adopted in this study is not validated for clinical use.
b) Beyond the small sample size and the retrospective design, also the choice of a data-driven TMB cutoff value could increase the risk of bias.
c) A discussion about co-mutations should be provided (see point 2)
d) Regarding 3rd point: “assessment on tumors harboring other genetic alterations, such as ROS1 and ALK fusions, was not made because of limited number of patients in our cohort”, I do not consider it a limitation of this study; it could be a different study exploring the outcome of a different population.
Reviewer 2 Report
The paper from Ji Youn Sung and coworkers reports the prognostic impact of TMB in advanced lung cancer patients with EGFR muutated adenocarcinoma treated wioth first/second line EGFR TKIi. The data show that a higher TMB is associated with worse parmeters (more liver mets, more involvede organs) and a worse prognosis (lower RR, shorter PFS and OS).
The data are clear, well reported and fairly discussed. The authors should however discuss more deeply the ther erason to choose their definition of low/high TMB: in the literature there is a general consensus to define a high TMB as more than 10 mutations for Mb. More data are also needed on salvage treatments: a low TMB was associated with a higher proportion of T790M patients; there was an imbalance in osimertinib treated patients according to olw vs high TMB? If yes this might account for a better PFS and OS reported for low TMB patients.
Reviewer 3 Report
Authors investigated the impact of TMB on EGFR-mutated NSCLC that was treated with EGFR-TKI. In spite of advance of therapy for this disease, the resistance after TKI treatment remained to be resolved. Their study is very interesting and attractive for shedding light on the critical problem of EGFR mutated NSCLC. I have 2 questions to the authors.
1. It is difficult to take tissue samples suitable for NGS from lung cancer. Please describe the detail methods for sampling.
2. As authors mentioned, TMB is the biomarker for ICIs therapy. Another question is the responsiveness to ICIs after the failure of TKI treatment in this cohort. Did TMB status affect efficacy of ICIs? Another biomarker for ICIs, PD-L1 staining. Was there any relationship between TMB and PD-L1 status?
Round 2
Reviewer 1 Report
Well done!